# The Relationship between Emotional Intelligence and Entrepreneurial Self-Efficacy of Chinese Vocational College Students

**DOI:** 10.3390/ijerph17124511

**Published:** 2020-06-23

**Authors:** Ya Wen, Huaruo Chen, Liman Pang, Xueying Gu

**Affiliations:** 1School of Education Science, Nanjing Normal University, Nanjing 210046, China; 170601026@stu.njnu.edu.cn (Y.W.); 190601021@stu.njnu.edu.cn (H.C.); 190602089@stu.njnu.edu.cn (L.P.); 2Center for Research and Reform in Education, Johns Hopkins University, Baltimore, MA 21286, USA

**Keywords:** Chinese vocational college students, relationship, emotional intelligence, entrepreneurial self-efficacy

## Abstract

The purpose of this study is to explore the relationship between emotional intelligence and entrepreneurial self-efficacy. The sample consisted of 529 students. The tools used to measure the relationship between emotional intelligence and entrepreneurial self-efficacy were the Emotional Intelligence Scale developed by Wong and Law and the Entrepreneurial Self-Efficacy Scale developed by Zhan. The results showed that there was a significant difference between male and female college students in entrepreneurial self-efficacy, but no significant difference between male and female college students in emotional intelligence. In entrepreneurial self-efficacy as well as emotional intelligence, there were significant differences between the third grade and the first and second grade, respectively. In addition, the results showed a significant positive correlation between entrepreneurial self-efficacy and emotional intelligence. With the improvement of the emotional intelligence level of vocational college students, the entrepreneurial self-efficacy will increase. The lower the emotional intelligence, the faster the improvement in entrepreneurial self-efficacy. The higher the emotional intelligence, the more stable the entrepreneurial self-efficacy. The university stage is considered an ideal entrepreneurial period, especially for vocational colleges’ students, who pay more attention to entrepreneurship and innovation education. Encouraging the cultivation of the emotional intelligence of vocational college students in life will help to improve personal entrepreneurial self-efficacy. This research emphasizes that improving the emotional intelligence of vocational college students can enhance their sense of self-efficacy in entrepreneurship and help students with entrepreneurship and career development.

## 1. Introduction

Entrepreneurship is of great significance to a country’s economic growth and decline in unemployment [1,2]. Under the policy background of “mass entrepreneurship and innovation” in China, the research on college students’ innovation and entrepreneurship has grown exponentially. How to enhance the self-efficacy of college students’ entrepreneurship and promote their entrepreneurship and employment to the maximum extent is an important issue for colleges to accelerate economic and social development, improve the talents cultivation pattern, and promote the success of college students [3].

Vocational colleges are an important part of China’s higher education system, vocational students are the guarantee of technical talents for China’s development [4]. Generally, they need to study in schools for three years [5]. In recent years, with the global economic recession, especially the global pandemic of COVID-19, the difficulty of employment has become an important challenge for Chinese vocational students. In recent years, China has attached great importance to the development of entrepreneurship education in vocational colleges. In such an environment, the entrepreneurship of vocational college students has received more attention.

Based on the theory of planned behavior, entrepreneurship is influenced by many factors, but entrepreneurial self-efficacy is emphasized as a key antecedent variable [6]. It is found that entrepreneurial self-efficacy is an important predictor of entrepreneurial intentions and behaviors. [7,8]. In recent years, emotional intelligence has become a research hotspot, scholars are increasingly interested in emotional intelligence research [9,10]. A large number of studies have shown that emotional intelligence is closely related to personal physical and mental health, subjective well-being, and can predict career adaptability and job satisfaction [11,12].

Although previous studies have explored the value of emotional intelligence for individual career success, few studies have analyzed the relationship between emotional intelligence and individual differences in entrepreneurship, such as the relationship between emotional intelligence and entrepreneurial self-efficacy [13,14].

It is one of the focuses of academic circles to explore the factors that affect the entrepreneurial self-efficacy of college students in vocational colleges. The purpose of this study is to explore the relationship between emotional intelligence and entrepreneurial self-efficacy of Chinese vocational college students. The remainder of this paper is organized as follows: Literature Review briefly reviews the literature of emotional intelligence and entrepreneurial self-efficacy; Materials and Methods describes the research participants, tools, data sources, and analysis methods; Results analyzes the research results; Discussion summarizes and discusses the results; Conclusion gives the conclusions of this paper, clarifies the impact of the research results on theory and practice, limitations of the research, and possible research directions in the future.

## 2. Literature Review

### 2.1. Emotional Intelligence

There are currently three models of research on emotional intelligence: Ability Model represented by Mayer and Salovey, Mixed Model represented by Bar-On, and Trait Model represented by Petrides [15,16]. In 1990, Mayer and Salovey proposed a new type of intelligence, called emotional intelligence, which refers to individuals accurately perceiving emotions, using emotions to accurately promote thinking, understanding emotions, and managing emotions of themselves and others [17,18]. Research by Mayer et al. (2016) shows that emotional intelligence is best measured by ability [19]. Bar-On proposed the following definition: “Emotional intelligence is a series of non-cognitive abilities and skills that influence an individual’s success in coping with environmental needs and stress” [20]. According to Petrides et al. (2016), trait emotional intelligence refers to an individual’s cognition of personal emotional ability, which refers to individual ability to understand, regulate, and express emotions to adapt to the environment and maintain health [21].

Since 2000, the literature on emotional intelligence has been endless, especially the research on positive psychological qualities such as emotional intelligence and well-being has attracted the attention of many scholars. The relationship between emotional intelligence and subjective well-being was analyzed, supporting emotional intelligence as a key skill for personal growth and social development [22]. There is a positive correlation between trait emotional intelligence and subjective well-being; a negative correlation between emotional intelligence and stress and social anxiety has also been confirmed [23]. Regarding the relationship between adolescent emotional intelligence and mental health, many researchers have conducted research and found that adolescents with lower parental violence have higher emotional intelligence, and emotional intelligence is related to positive adolescent development [24,25,26]. In addition, studies have found that adolescents with low emotional intelligence report a higher risk of suicide than those with high emotional intelligence [27]. A study of 524 Italian college students found that traits of emotional intelligence are a source of happiness and hope [28].

In the field of organizational behavior, emotional intelligence is also concerned by scholars. A study shows that employee emotional intelligence has a positive predictive effect on psychological capital and job performance, and some studies have concluded that the interaction between role ambiguity and emotional intelligence has significant significance in explaining the dimension of engagement [29,30,31]. Career adaptability is a popular concept in organizational behavior in recent years [32,33]. A study based on cross-lagged panel analysis found that emotional intelligence can predict career adaptability [34]. In addition, a study of senior managers showed that the relationship between emotional intelligence and occupational personality scales showed that emotional intelligence was positively correlated with many occupational scales [35].

Can emotional intelligence be cultivated? How effective is the cultivation? Many researchers have discussed this. A meta-analysis of emotional intelligence training shows that, regardless of the form, training interventions for emotional intelligence are considered effective [36,37]. In the field of organizational behavior, studies have found that interventions on emotional intelligence may effectively improve job satisfaction [38]. An interview-based study found that training nurses’ emotional intelligence is beneficial to their mental health [39].

### 2.2. Entrepreneurial Self-Efficacy

When it comes to entrepreneurial self-efficacy, self-efficacy needs to be discussed. The concept of self-efficacy is derived from Bandura’s social learning theory. According to the basic perspective of social learning theory, generally speaking, when people are in or facing a predicament that is bad for themselves, many people often show psychological fear and try to avoid or get rid of various unfavorable situations and problems that they think are difficult to cope with [40,41]. The self-efficacy refers to the ability of individuals to show very decisive judgments and behaviors when they are in or facing such unfavorable situations or problems, to effectively complete tasks, and to overcome difficulties and problems [42,43]. Hackett and Betz (1981) proposed to extend the theory of self-efficacy to the career field. At the same time, some researchers applied self-efficacy to the field of entrepreneurship to generate entrepreneurial self-efficacy [44,45]. Some scholars describe entrepreneurial self-efficacy as entrepreneurial self-confidence in specific tasks [46]; others define entrepreneurial self-efficacy as confidence in individuals’ ability to complete the entrepreneurial process [47]. Some scholars defined the concept of entrepreneurial self-efficacy from three dimensions: the first dimension is to apply self-efficacy to specific aspects of entrepreneurial spirit; the second dimension is to emphasize the content level of self-efficacy, the third dimension is the validity of self-efficacy belief [48].

Entrepreneurial self-efficacy is an important influencing factor of entrepreneurial intention, and entrepreneurial intention is a key indicator of entrepreneurial behavior prediction and interpretation [49]. The relationship between entrepreneurial self-efficacy and entrepreneurial intention has attracted the attention of many scholars, especially in the field of higher education. Some researchers have found that there is a positive correlation between the quality of entrepreneurship education and entrepreneurial self-efficacy [50]. In practice-oriented entrepreneurship courses, higher entrepreneurial self-efficacy is associated with higher entrepreneurial intention [51], and entrepreneurial self-efficacy has a strong predictive effect on entrepreneurial intention [52]. In addition, some researches based on the Social Career Cognition Theory (SCCT) also show that entrepreneurial self-efficacy is positively related to entrepreneurial intention [53]. Some scholars have pointed out that entrepreneurial self-efficacy has a strong impact on entrepreneurial self-efficacy for college students [54]. A study of engineering college students found that, in addition to the positive moderation effect of social norms on the relationship between entrepreneurial self-efficacy and entrepreneurial intention, entrepreneurial education is also positively related to the intention of entrepreneurial activities [55]. Other researchers have found that the relationship between employment perception, entrepreneurial intention, career adaptability, and self-efficacy of college students and job seekers is positively correlated with career adaptability and general self-efficacy [56]. In summary, entrepreneurial self-efficacy is an important factor affecting entrepreneurial intention, and it is of great significance for individuals to form entrepreneurial intention or to complete entrepreneurial behaviors.

Entrepreneurial self-efficacy is closely related to the performance of enterprises. A study on the self-efficacy of entrepreneurs in Central Asia concluded that self-efficacy has a direct and intermediary effect on performance [57]. In addition, the evidence on South African enterprises supports that entrepreneurial self-efficacy is significantly related to the competitiveness of enterprises during the entrepreneurial stage of searching, planning, and integrating resources and personnel [58]. Besides, a study on the performance of small French companies has concluded that self-efficacy and work efficiency are positively related to corporate performance [59]. Based on multilevel regression analyses, it was found that entrepreneurial self-efficacy can be used as a personal resource to help entrepreneurs turn increasing uncertainty into exploration and opportunity identification [60].

Studies by some scholars have shown that entrepreneurial self-efficacy is an important variable in predicting entrepreneurial behavior and entrepreneurial success in the field of entrepreneurship. Entrepreneurial self-efficacy is affected by individual internal and external factors, including social, cultural, and economic background, personality, and ability. Entrepreneurial motivation, entrepreneurial attention, and participation are extremely significantly correlated with entrepreneurial self-efficacy [61]. In recent years, a large number of studies have shown that entrepreneurial self-efficacy, as an important part of entrepreneurial cognition, is significantly related to entrepreneurial motivation and ability, and has a good predictive effect on entrepreneurial decision-making, behavior, and performance [62].

In addition, the relationship between entrepreneurial self-efficacy and some demographic variables, such as gender, has also attracted the attention of researchers. As early as 2011, some researchers have explored the role models and self-efficacy on the formation of career intentions, and found that role models have a strong effect on females’ self-efficacy [63,64]. Some scholars have explored the relationship between the sample’s gender role and the entrepreneur’s self-efficacy under the cross-cultural background, and the research shows that the entrepreneur’s self-efficacy is more affected by the gender role positioning [65]. In addition, a study on entrepreneurial self-efficacy of college students showed that female college students’ self-efficacy and the number of entrepreneurial role models were related to increased entrepreneurial intention [66].

### 2.3. Emotional Intelligence and Entrepreneurial Self-Efficacy

At present, there are relatively few studies on the relationship between emotional intelligence and entrepreneurial self-efficacy. Some studies have found that entrepreneurial passion has a mediating role in the relationship between self-efficacy and sustainability, indicating that emotion has an important value in entrepreneurship [67]; based on the Fuzzy-Set Qualitative Comparative Analysis (FSQCA), some researchers conducted causal and effective decision tests on the structural effects of entrepreneurial passion, entrepreneurial self-efficacy, and risk perception [68].

Some studies have explored the relationship between emotional intelligence and entrepreneurial self-efficacy. A survey based on Britons (16–84 years) showed that the differences in individual entrepreneurship were partly caused by differences in trait emotional intelligence [14]. In addition, some scholars have researched the relationship between entrepreneurial self-efficacy of Spanish entrepreneurs and college students and their Big Five personality and emotional intelligence, and found that entrepreneurs and students with high emotional intelligence have common entrepreneurial psychological characteristics, such as extraversion, openness, high emotional intelligence score, and low neuroticism score [69]. A study showed that there is a significant difference in the entrepreneurial ability of college students with different emotional intelligence levels, and emotional intelligence has a strong predictive power for entrepreneurial ability [70].

Emotional intelligence has a good predictive effect on college students’ entrepreneurial self-efficacy [71]. Vocational colleges are one of the important types of higher education in China, and they are an important part of vocational education. At this stage, college students and undergraduates of vocational colleges belong to the same age group and are comparable and can be used for reference at the same level. From the theoretical perspective, the study of the relationship between entrepreneurial self-efficacy and emotional intelligence of Chinese vocational college students is helpful to explore and improve entrepreneurial theory, test the cross-cultural consistency of entrepreneurial theory, and at the same time, expand the research object of emotional intelligence and entrepreneurial self-efficacy. From the perspective of practice, in recent years, China has increasingly emphasized the role of vocational education in the development of social economy. It is of positive value to study the emotional intelligence and entrepreneurial self-efficacy of vocational college students for the development of entrepreneurial education. However, there are few studies on the emotional intelligence and entrepreneurial self-efficacy of students in Chinese vocational colleges.

## 3. Materials and Methods

### 3.1. Participants

In this study, questionnaires were distributed in some provinces of China. Among the students of Grade 1–3 in liberal arts, science and engineering, and management in more than 10 vocational colleges, a total of 550 students were randomly selected as the objects of investigation. A total of 529 valid questionnaires were collected, with an effective rate of 96%. Among them, 290 are male, accounting for 55%; 239 are female, accounting for 45%. There are 196 students in the first grade, 157 in the second grade, and 176 in the third grade.

### 3.2. Instruments

#### 3.2.1. Entrepreneurial Self-Efficacy Scale

Using the *Entrepreneurial Self-Efficacy Scale* (ESES) compiled by Zhan. The scale has a total of 19 items, which are composed of four factors of opportunity recognition efficacy, relationship efficacy, management efficacy, and risk tolerance efficacy [72]. The basic information of ESES is shown in Table 1, and details of topics can be seen in Appendix A
Table A1. Using Likert’s seven-point scoring method, the higher the score, the higher the level of entrepreneurial self-efficacy in the project. The Cronbach’s alpha for this study was 0.946.

#### 3.2.2. Emotional Intelligence Scale

Emotional intelligence was measured using *Wong and Law’s “Emotional Intelligence Scale”* (WLEIS), the scale has a total of 16 items, including four dimensions of appraisal of self-emotions, appraisal of others’ emotions, regulation of emotion, and use of emotion on cognition [73]. The basic information of WLEIS is shown in Table 2, and details of topics can be seen in Appendix A
Table A2. Using Likert’s seven-point scoring method, the higher the score, the higher the emotional intelligence in the project. The Cronbach’s alpha for this study was 0.936.

### 3.3. Procedure

Data collection was in the form of paper quality forms distributed in class. First, the education center was contacted, the purpose of the study explained, and authorization requested to complete the scale. The answer of the scale is anonymous, the data filled in by students is confidential, and the research purpose is exclusive. In this study, the distribution, filling, and recovery of the scale were conducted during the class time; the students who participated in filling in the scale were in a suitable environment without external interference for about 30 min.

### 3.4. Data Processing

All data were analyzed by SPSS 25 (IBM, New York, NY, USA) and Amos 7.0 (IBM, New York, NY, USA), including descriptive statistics, variance analysis, correlation analysis, and regression analysis.

## 4. Results

### 4.1. General Description

The basic information of the research sample is shown in Table 3. Among the vocational college students participating in this study, 37% were first grade, 30% were second grade, and 33% were third grade. Of the students, 86% had “work experience” and 14% “no work experience”. Of the students, 40% had never had entrepreneurship training, 50% of students had attended one or two entrepreneurial courses or lectures, and 10% of students had received more systematic entrepreneurship education.

Table 4 shows that in terms of entrepreneurial self-efficacy, the overall performance of vocational college students shows a positive trend. The highest score is 133, the lowest is 19, and the median is 99. If the score exceeds the median, the entrepreneurial self-efficacy tends to be positive. In the total scale and scores, the opportunity recognition, relationship, and management mean values of vocational college students’ entrepreneurial self-efficacy exceed the corresponding median value, but the overall mean value and risk tolerance are slightly lower than the median value, which shows that the current college students’ entrepreneurial self-efficacy tends to be positive, but the overall level is only in the middle level. Among them, the average score of opportunity recognition is the highest, followed by relationship, management, and risk tolerance.

Table 4 shows that in terms of emotional intelligence, the overall trend of vocational college students also shows a positive trend. The highest score of total emotional intelligence scale is 126, the lowest score is 37, and the median score is 85. If the score exceeds the median, it can be considered that the emotional intelligence tends to be positive. In the total scale and scores, the average value of emotional intelligence of vocational college students is higher than the corresponding median value, but only slightly higher than the median value, which shows that the current college students’ emotional intelligence tends to be positive, but the overall level is only above the middle level. Among them, the average score of use of emotion is the highest, followed by regulation of emotion, appraisal of others’ emotions, and appraisal of self-emotions.

### 4.2. Correlation Description

In order to analyze the correlation between emotional intelligence and entrepreneurial self-efficacy. According to Pearson product–moment correlation, there is a correlation between emotional intelligence and entrepreneurial self-efficacy. The correlation coefficient between the two variables is 0.413, which belongs to a moderate degree of correlation. Cronbach’s alpha reliability values for the scales used are given in Table 5 on the diagonals in bold.

### 4.3. Model Fitness, Effectiveness, and Reliability

CFA was used to test whether the measurement model is consistent with the data, and several fitting indexes were calculated to determine whether the structural model is consistent with the sample data. All values obtained from the model fitting index indicate that the model agrees well with the data. More specifically, x^2^ = 772.176, df = 524, x^2^/df = 1.474, Goodness of Fit Index (GFI) = 0.892, Adjusted Goodness of Fit Index (AGFI) = 0.877, Tucker–Lewis Index (TLI) = 0.970, Norm Fit Index (NFI) = 0.919, the Comparative Fit Index (CFI) = 0.972, and the Root Mean Square Error of Approximation (RMSEA) = 0.035, indicating that the model fits well. The CFA estimates indicate that the proposed four-factor model is best suited to current sample data. All items load well on their respective factors, with factor loading greater than 0.5. Table 6 shows the Average Variance Extraction (AVE), Composite Reliability (CR), Maximum Shared Variance (MSV), and average square root value. The CR value greater than the specified threshold of 0.7 indicates high reliability. In all of our study structures, the value of AVE is greater than 0.5, which indicates a high level of convergent validity. For all variables used in the study, the square root of AVE is greater than the inter-structure correlation, which indicates a high level of discriminant validity. When the MSV value is less than the AVE value, discriminant validity is further established. The results show that all MSV values are less than AVE, which provides further support for discriminant validity.

### 4.4. Gender Differences in Entrepreneurial Self-efficacy and Emotional Intelligence

The scores of entrepreneurial self-efficacy and emotional intelligence of college students of different genders were tested by independent sample t-test (see Table 7). The results showed that under the condition of homogeneity of variance, there was significant difference between male and female college students in entrepreneurial self-efficacy (t = 3.933, *p* < 0.001). In the four subscales, opportunity recognition, relationship, and risk tolerance of male college students were significantly higher than that of female college students, and management was not significant. There was no significant difference in emotional intelligence between male and female college students (t = 2.553, *p* > 0.05). Among the four subscales, only appraisal of others’ emotions and regulation of emotion are significant.

### 4.5. Grade Differences in Entrepreneurial Self-efficacy and Emotional Intelligence

The scores of entrepreneurial self-efficacy and emotional intelligence of college students in different grades were tested by independent sample t-test (see Table 8 and Table 9). The results showed that under the condition of homogeneity of variance, there was no significant difference in entrepreneurial self-efficacy of college students in grade one, two, and three. In order to explore the specific sources of differences, LSD multiple comparative analysis was carried out. The results showed that there were significant differences between the third grade and the first and second grade, respectively. There was no difference in the opportunity dimension of entrepreneurial self-efficacy among different grades. There was no significant difference in the other dimensions of entrepreneurial self-efficacy between grade one and grade two. There was significant difference in the other dimensions of entrepreneurial self-efficacy between grade one and grade three and grade two and grade three.

The results show that, in terms of emotional intelligence, there was no significant difference in entrepreneurial self-efficacy between grade one, grade two, and grade three students. In order to explore the specific sources of differences, LSD multiple comparative analysis was carried out. The results showed that there were significant differences between the emotional intelligence of grade three and grade one and grade two, respectively; there was no significant difference between the emotional intelligence of grade one and grade two college students.

### 4.6. Regression Analysis

A detailed analysis of the regression of entrepreneurial self-efficacy according to different emotional intelligence dimensions shows that there is a linear relationship between entrepreneurial self-efficacy and emotional intelligence of vocational college students, as shown in Figure 1. At the same time, based on the data, according to the research of Guerra Bustamante et al., this paper divides emotional intelligence into three levels of low, middle, and high according to the standard of percentile, and makes linear regression analysis [74]. On the basis of the regression equation of emotional intelligence and entrepreneurial self-efficacy, especially the state of the slope, we found that when emotional intelligence increases, individuals will think that their entrepreneurial self-efficacy is higher, and when emotional intelligence decreases, individuals have less entrepreneurial self-efficacy. In addition, we also found that when emotional intelligence is low, entrepreneurial self-efficacy improves faster than when emotional intelligence is higher.

## 5. Discussion

By exploring the emotional intelligence and entrepreneurial self-efficacy of Chinese vocational college students, this study expanded the previous literature research concerning emotional intelligence and entrepreneurial self-efficacy of vocational college students. This study explores the entrepreneurial self-efficacy and emotional intelligence of a sample of Chinese vocational college students. On the one hand, it broadens the field of emotional intelligence; on the other hand, it explores the factors that affect entrepreneurial self-efficacy. First, our research showed that there is a positive correlation between entrepreneurial self-efficacy and emotional intelligence reported by vocational college students. We also found that entrepreneurial self-efficacy increased most quickly when emotional intelligence was low, and entrepreneurial self-efficacy was more stable when emotional intelligence was higher.

### 5.1. An Analysis of Gender Differences in Entrepreneurial Self-Efficacy and Emotional Intelligence

We found that the total score of male entrepreneurial self-efficacy of vocational college students was significantly higher than that of female students, which shows that from the overall point of view, male students are more confident in entrepreneurship than female students. Similar results have been obtained in other studies. Zhou and Yang, taking undergraduates and postgraduates as samples, found that male scores in entrepreneurial self-efficacy were significantly higher than female scores [75]. Jin’s research on college students also found that male students scored significantly higher than female students in all dimensions of entrepreneurial self-efficacy [76]. Specifically, opportunities recognition, relationship, and risk tolerance of male college students were significantly higher than that of female college students, and management was not significant, which may be related to the impact of traditional Chinese culture on women. For vocational college students, the difference in the management was not significant, which is also a noteworthy discovery.

This study found that there was no significant difference in emotional intelligence between male and female students in vocational colleges. Previous studies have shown that the difference between men and women in emotional intelligence has been inconclusive. Specifically, there were significant differences between male and female college students in appraisal of others’ emotions and regulation of emotion. Satsangi and Agarwal et al. found that there was significant gender difference in emotional intelligence of postgraduates, and female postgraduates reported higher emotional intelligence scores [77]. Zhao et al. showed that there was no significant gender difference in emotional intelligence of Chinese college students [78]. Marzuki et al. did not find gender differences in emotional intelligence among university students [79]. On the one hand, the gender difference of emotional intelligence of vocational college students may be related to the scale selected for measurement; on the other hand, although the same scale may have an impact on the measurement results, such as the number of subjects sampled, region, nationality, and so on, it may be possible to find the reasons from the fields of biology, psychology, sociology, education, and so forth.

### 5.2. An Analysis of Grade Differences in Entrepreneurial Self-Efficacy and Emotional Intelligence

In this study, the comparison of entrepreneurial self-efficacy among the three grades found that the entrepreneurial self-efficacy of the third grade students was significantly higher than that of the first and second grade students, and there was no significant difference between the first and second grade students. This is different from some studies. For example, Chen and Yin’s research showed that there is no significant difference in entrepreneurial self-efficacy among college students of different grades [80].The result of this study may be that the college students in vocational colleges need to find jobs in the third grade, and it may be more difficult for the college students in vocational colleges to find jobs than the general four-year undergraduate students in China. In this context, entrepreneurship, as an important employment path, may be actively concerned by the graduates in the third grade. Therefore, the third grade students have a higher sense of entrepreneurship self-efficacy.

On the comparison of emotional intelligence of three grades of students in vocational colleges, it was found that the emotional intelligence of the third grade students was significantly higher than that of the first and second grade students, and there was no significant difference between the first and second grade students, which is different from the previous research. Lu et al. investigated the emotional intelligence of vocational college students and found that the emotional intelligence of junior students was lower than that of freshmen and sophomores; Zhang and Xu also found that there was a period of low emotional intelligence in the junior students [81,82]. The emotional intelligence of the junior school students in this study was higher than that of the students in other grades, probably because with the growth of individual age comes the enrichment of life experience, the increase of college students’ maturity, the improvement of their perception of self and other emotions, and the more effective management and regulation of their own emotions [83,84].

### 5.3. Relationship between Entrepreneurial Self-Efficacy and Emotional Intelligence

As in previous studies, we found that there was a significant positive correlation between entrepreneurial self-efficacy and emotional intelligence, indicating that emotional intelligence is one of the influencing factors of entrepreneurial self-efficacy. People with high emotional intelligence also have strong entrepreneurial self-efficacy, which may be due to the importance of self-perception and self-regulation in the development of entrepreneurial self-efficacy, while emotional intelligence involves emotional evaluation and regulation, which may promote the improvement of entrepreneurial self-efficacy [85].

In addition, the results showed that emotional intelligence had a significant negative correlation with gender, a significant positive correlation with grade, and a significant negative correlation with major. There was a significant negative correlation between entrepreneurial self-efficacy and gender, a significant positive correlation between entrepreneurial self-efficacy and grade, and a significant negative correlation between entrepreneurial self-efficacy and major.

In conclusion, the correlation between entrepreneurial self-efficacy and emotional intelligence was high, and gender, age and other variables had little relationship with entrepreneurial self-efficacy and emotional intelligence. The correlation coefficient between entrepreneurial self-efficacy and emotional intelligence was large, indicating that college students with high emotional intelligence also have a stronger entrepreneurial self-efficacy. The entrepreneurial process means the establishment and maintenance of multiple interpersonal relationships, which means more competition and pressure. Managing and regulating one’s emotions is an essential quality for successful entrepreneurs [12,86,87]. Only when entrepreneurs have the ability to control and adjust their own emotions, and know how to think in a different way, can they form a higher sense of self-efficacy, and have the confidence to recognize entrepreneurial opportunities in the environment, conduct interpersonal relationship management, entrepreneurial management, and tolerate entrepreneurial risks [88,89,90].

### 5.4. Relationship between Different Levels of Emotional Intelligence and Entrepreneurial Self-Efficacy

At the same time, we found that when emotional intelligence is at a lower level, the improvement of entrepreneurial self-efficacy is more obvious; while when emotional intelligence is at a higher level, the improvement of entrepreneurial self-efficacy is relatively slow. This seems to indicate that when the emotional intelligence of vocational college students is low, improving the individual’s emotional intelligence is more likely to improve students’ entrepreneurial self-efficacy. However, when the level of emotional intelligence of college students is high, the improvement of entrepreneurial self-efficacy becomes relatively slow. Does this mean that at this stage, in addition to emotional intelligence having a certain predictive effect on entrepreneurial self-efficacy, other factors may also have some impact on entrepreneurial self-efficacy, such as some personality characteristics, psychological qualities of the individual, or some other external factors [91,92,93].

## 6. Limitations

There are some limitations when explaining current research. First, the cross-sectional design was used in this study, and the relationship between the variables cannot be explained causally. From this perspective, longitudinal research is more likely to clarify the relationship between entrepreneurial self-efficacy and emotional intelligence of vocational college students [94]. Moreover, the sample size selected in this study was limited. Perhaps a larger sample size will make the effect of this study more obvious [95]. The findings of this study require more college students from vocational colleges to participate in repeated tests before they are more likely to increase the universality of our results.

## 7. Implications for Further Research

Although this study has some limitations, this study also has its unique value. Previous studies have studied the factors that affect entrepreneurial self-efficacy, but there is less research on the relationship between emotional intelligence and entrepreneurial self-efficacy. Besides, an interesting finding of this study is that individuals with different emotional intelligence levels have significant differences in entrepreneurial self-efficacy, which is also a complement to previous research.

This research is not only of special significance to the entrepreneurship education of Chinese vocational college students, but also may play a positive role in the entrepreneurship education of other technical students with similar background of higher education in the world. In the 21st century of the knowledge economy, the successful entrepreneurship of an individual is inseparable from a high sense of entrepreneurial self-efficacy. It is particularly important to carry out entrepreneurship education at the university stage to improve the individual’s entrepreneurial self-efficacy [96,97,98]. For example, in the design of undergraduate entrepreneurship education, emotional intelligence is included, and social and emotional learning courses are referenced to comprehensively improve the emotional intelligence of college students in vocational colleges in China, helping to improve college students’ entrepreneurial self-efficacy [99,100,101,102]. In addition, future research also needs to consider other factors, such as entrepreneurial personality and entrepreneurial passion, which affect vocational college students’ entrepreneurial self-efficacy, in addition to emotional intelligence. Finally, on the basis of on a comprehensive exploration of the factors that affect entrepreneurial self-efficacy, emotional intelligence and other factors are incorporated into the system of entrepreneurship education for vocational college students. Through the combination of multiple factors, the improvement of entrepreneurial self-efficacy can be promoted among college students, thus increasing the probability of successful entrepreneurial behavior, creating the entrepreneurial environment for Chinese vocational college students and other similar higher-education-system students in the world, improving the employment quality of college students, and promoting the social and economic development.

## 8. Conclusions

In summary, it is important to study the relationship between entrepreneurial self-efficacy and emotional intelligence of college students in Chinese vocational colleges. In order to further enhance the entrepreneurial self-efficacy of college students in vocational colleges, the future research can try the following three aspects: first, improve the educational training path for college students’ emotional intelligence; second, explore related factors that affect entrepreneurial self-efficacy in addition to emotional intelligence and explore the common mechanism of emotional intelligence and other factors on the entrepreneurial self-efficacy of vocational college students; third, explore the influence of emotional intelligence, entrepreneurial self-efficacy, and other variables on entrepreneurial intention of students in Chinese vocational colleges.

## Figures and Tables

**Figure 1 ijerph-17-04511-f001:**
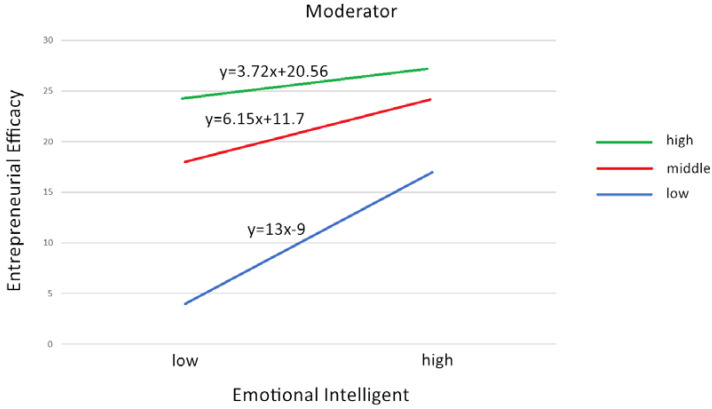
Relationship between different levels of emotional intelligence and entrepreneurial self-efficacy.

**Table 1 ijerph-17-04511-t001:** Basic information of entrepreneurial self-efficacy scale.

Dimension	Items (No Reverse Questions)
Opportunity recognition	1, 2, 3, 4
Relationship	5, 6, 7, 8, 9
Management	10, 11, 12, 13, 14
Risk tolerance	15, 16, 17, 18, 19

**Table 2 ijerph-17-04511-t002:** Basic information of emotional intelligence scale.

Dimension	Items (No Reverse Questions)
Appraisal of self-emotions	1, 2, 3, 4
Appraisal of others’ emotions	5, 6, 7, 8
Regulation of emotion	9, 10, 11, 12
Use of emotion on cognition	13, 14, 15, 16

**Table 3 ijerph-17-04511-t003:** Basic information of the sample.

Variable	Category	N	%
Gender	Male	290	0.55
	Female	239	0.45
Grade	First grade	196	0.37
	Second grade	157	0.30
	Third grade	176	0.33
Work experience	Have	454	0.86
	No	75	0.14
Attend training	Never had	213	0.40
	Listened to one or two entrepreneurial courses or lectures	264	0.50
	Received a more systematic entrepreneurship education	52	0.10

**Table 4 ijerph-17-04511-t004:** Correlation.

Scale	Dimension	M	SD	Cronbach a
Entrepreneurial Self-Efficacy	**ESE-Total**	98.921	16.848	0.951
	Opportunity recognition	22.075	4.044	0.713
	Relationship	21.323	3.631	0.805
	Management	21.175	3.788	0.845
	Risk tolerance	20.974	3.787	0.840
Emotional Intelligence	**EI-Total**	85.547	12.086	0.896
	Appraisal of self-emotions	20.331	4.295	0.895
	Appraisal of others’ emotions	26.207	4.766	0.858
	Regulation of emotion	26.106	5.157	0.898
	Use of emotion on cognition	26.278	5.086	0.894

**Table 5 ijerph-17-04511-t005:** Correlation description between entrepreneurial self-efficacy scale and emotional intelligence scale.

Variable	Mean	SD	1	2	3	4	5
**1. Sex**	1.45	0.50	-				
**2. Grade**	1.96	0.84	0.091 *	-			
**3. Major**	1.41	0.56	0.705 **	0.110 *	-		
**4. EI**	85.547	12.086	−0.126 **	0.174 **	−0.097 *	**0.896**	
**5. ESE**	98.921	16.848	−0.123 **	0.223 **	−0.096 *	0.413 **	**0.951**

Note: EI: Emotional Intelligence; ESE: Entrepreneurial Self-Efficacy; *: *p* < 0.05; **: *p* < 0.01; the bold: self-correlation.

**Table 6 ijerph-17-04511-t006:** Composite reliability, maximum shared variance, average variance extraction, and square roots of AVE.

Variable	CR	AVE	MSV	Square Roots of AVE
Entrepreneurial Self-Efficacy	0.946	5.19	1.26	2.278
Emotional Intelligence	0.936	5.31	1.23	2.304

Note: CR = Composite Reliability; AVE = Average Variance Extraction; MSV = Maximum Shared Variance.

**Table 7 ijerph-17-04511-t007:** Gender Differences.

	Male		Female		t
	M	SD	M	SD	
**Entrepreneurial Self-Efficacy**	101.435	14.273	95.577	19.291	3.933 ***
Opportunity recognition	20.852	3.726	19.637	4.874	3.182 ***
Relationship	26.800	3.975	25.417	5.562	3.267 ***
Management	26.772	4.811	25.220	5.469	3.393
Risk tolerance	27.010	4.621	25.302	5.506	3.795 **
**Emotional Intelligence**	86.727	11.204	83.977	13.028	2.553
Appraisal of self-emotions	21.941	3.676	22.252	4.490	−0.857
Appraisal of others’ emotions	21.685	3.250	20.876	4.047	2.416 *
Regulation of emotion	21.559	3.522	20.665	4.067	2.647 **
Use of emotion on cognition	21.569	3.574	20.183	3.922	4.146

Note: * *p* < 0.05, ** *p* < 0.01, *** *p* < 0.001.

**Table 8 ijerph-17-04511-t008:** Grade differences in entrepreneurial self-efficacy.

		SS	df	MS	F	Sig
**Entrepreneurial Self-Efficacy**	Intergroup	4481.340	2	2240.670	8.115	0.000
	Intragroup	139,429.511	505	276.098		
Opportunity recognition	Intergroup	61.726	2	30.863	1.678	0.188
	Intragroup	9290.715	505	18.397		
Relationship	Intergroup	389.888	2	194.944	8.847	0.000
	Intragroup	11,127.409	505	22.034		
Management	Intergroup	378.112	2	189.056	7.286	0.001
	Intragroup	13,104.148	505	25.949		
Risk tolerance	Intergroup	456.819	2	228.410	9.113	0.000
	Intragroup	12,657.045	505	25.063		

Note: SS: Sum of Squares; MS: Mean Square.

**Table 9 ijerph-17-04511-t009:** Grade differences in emotional intelligence.

		SS	df	MS	F	Sig
**Emotional Intelligence**	Intergroup	4409.697	2	2204.849	15.987	0.000
Intragroup	69,646.169	505	137.913		
Appraisal of self-emotions	Intergroup	422.018	2	211.009	13.538	0.000
Intragroup	7871.139	505	15.586		
Appraisal of others’ emotions	Intergroup	394.206	2	197.103	15.828	0.000
Intragroup	6288.849	505	12.453		
Regulation of emotion	Intergroup	159.324	2	79.662	5.653	0.004
Intragroup	7116.083	505	14.091		
Use of emotion on cognition	Intergroup	191.640	2	95.820	6.836	0.001
Intragroup	7079.027	505	14.018

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
