# Peer review of "The Relationship between Emotional Intelligence and Entrepreneurial Self-Efficacy of Chinese Vocational College Students"

_ijerph, 2020, doi:10.3390/ijerph17124511_

Round 1

Reviewer 1 Report

I appreciate the changes made to the final version of the document. The revision work done was very productive, it is well justified and, above all, it improved the final quality of the manuscript.

Author Response

Dear Reviewer:

We would like to thank you for the careful reading of our manuscript and for providing us with helpful comments to improve its quality.

Once again, thank you again for your valuable and constructive suggestions for this article. Your suggestions are very helpful for the improvement of the article. 

Kind regards,

Authors

Reviewer 2 Report

Dear Authors,
It is a pleasure to inform you that in my opinion the paper has been drastically improved after the first round of review. All comments on the first version of the paper have been followed and I think that the improvements are visible in all sections of the paper. In short, I can say that the paper offers new and significant information to justify the publication.
I only have a few marginal comments and minor suggestions. I hope these comments will be useful to improve the paper.

On page 2 authors say: “A large number of studies have shown that emotional intelligence is closely related to personal physical and mental health, subjective well-being, etc., and can predict career adaptability, job satisfaction, etc.”.

Probably the term “etc.” is not adequate in the introduction. A different form of the sentence could be more appropriate.

On page 3, authors say: “Emotional intelligence has received attention in many areas”. You could specify what are these areas.

Authors report limitations and implication as two sub-paragraphs of the paragraph “Conclusion”. Probably this is not completely correct. For this reason, I suggest to have to separate paragraphs (6. Limitations of research, 7. Implications for Further Research).

Please read carefully the text. There are some incomplete sentences. For example, on page 10: “Specifically,Appraisal of others’ emotionsRegulation of emotion”.

Finally, I have a doubt about the generalizability of the results. you have put this as limitation of the research. However, as ijerph is an international Journal, I think it is necessary to discuss why and how these results can be useful to an international audience.

Good luck with this research.

Author Response

Dear Reviewer:   

We would like to thank you for the careful reading of our manuscript and for providing us with helpful comments to improve its quality. In the revised version of the manuscript we have tried to address all the comments made by the reviewers. Please, find below a point-by-point response to your comments.

1.Thank you very much for your advice. We agree with you a hundred percent. Like you, we don't think it's appropriate to use "etc.". We adjust this part as follows: A large number of studies have shown that emotional intelligence is closely related to personal physical and mental health, subjective well-being, and can predict career adaptability, job satisfaction.

2.Thank you for this constructive suggestion. Your suggestion is very good, which is very helpful to improve the article. We agree with you and think that this sentence is not appropriate here. In view of the previous discussion on the application of emotional intelligence, this sentence will be deleted and will not be repeated. Thank you again for your detailed suggestions!

3.Your suggestion is very good! Thank you very much for your warm remind. According to your suggestion, we have divided the original sixth part into 6. Limitations of research and 7. Implications for further research. Thank you for your kind advices.

4.Thank you for your detailed suggestions. According to your suggestion, we have modified these parts: there are significant differences between male and female college students in appraisal of others’ emotions and regulation of emotion. Thank you.

5.We agree with you that when writing articles, we still need to consider the internationalization of journals. According to your constructive suggestions, we have made some modifications to the article, as follows:

Our research is not only of special significance to the entrepreneurship education of Chinese vocational college students, but also may play a positive role in the entrepreneurship education of other technical students with similar background of higher education in the world.

Through the combination of multiple factors, we can jointly promote the improvement of entrepreneurial self-efficacy among college students, thus increasing the probability of successful entrepreneurial behavior, creating the entrepreneurial environment for Chinese vocational college students and other similar higher education system students in the world, improving the employment quality of college students, and promoting the social and economic development.

Once again, thank you again for your valuable and constructive suggestions for this article. Your suggestions are very helpful for the improvement of the article. 

Kind regards

Authors

Reviewer 3 Report

The article has been improved and now, in my opinion, it deserves publication in the journal.

Author Response

Dear Reviewer:

We would like to thank you for the careful reading of our manuscript and for providing us with helpful comments to improve its quality.

Once again, thank you again for your valuable and constructive suggestions for this article. Your suggestions are very helpful for the improvement of the article. 

Kind regards,

Authors

This manuscript is a resubmission of an earlier submission. The following is a list of the peer review reports and author responses from that submission.

Round 1

Reviewer 1 Report

Improvement suggestions:

- The abstract emphasizes the gender of the sample. This control variable is not more relevant than others. Therefore, it shouldn’t be emphasized.

- I recommend dividing the introduction section into two sections. One with the introduction and the other with the literature review.

- The last paragraph of the introduction section should present the structure of the manuscript.

- Line 184: Kickul et al. citation doesn’t present the year.

- Authors should describe the concept of a vocational college. This concept is different in some European countries and USA.

- It is not clear what means the items column in Table 1 and Table 2.

- Authors should present the performed tests evidencing that no normality or homomorphism was found in the data.

- Entrepreneurial experience is mapped into three categories. This view is too simplified and it is not clear whether the entrepreneur created one or more businesses.

- It is not clear what is a more systematic entrepreneurship education.

- The division into three categories (e.g., low, medium, and high) is highly debatable. The "medium" category has a higher number of frequencies than the other categories.

- Authors should use and compare other methods to analyze the data like the multiple hierarchical regression analyses.

- The discussion of the results is rather weak and does not allow a comprehensive discussion of the study results.

- Limitations of study and implications for further research should be migrated to the conclusions section.

- The conclusions section should present the theoretical and practical contributions of this study.

- Proofreading is needed. Several mistakes were identified along the manuscript.

- Some relevant references were not mentioned, respectively:

https://www.krishisanskriti.org/vol_image/22Oct201505100654%20%20%20%20%20%20%20Monika%20Agarwal%20%20%20%20%20%20%20%20%20%201272-1275.pdf

https://www.sciencedirect.com/science/article/pii/S1576596214000164

https://www.tandfonline.com/doi/abs/10.1080/19420676.2017.1371628

https://articlegateway.com/index.php/JHETP/article/view/2533

Reviewer 2 Report

The paper explores the relationship between emotional intelligence and the level of entrepreneurial self-efficacy among Chinese vocational college students. Despite the purpose seems to be quite interesting, the paper has several limitations in theoretical and practical sections.

Introduction

Starting from the introduction, the section should be better framed and written. In the current version, the paragraph is very poor and it does not provide enough information about the research topic, the reference context and the gap that the work wants to fill.

The function of the introduction is to describe the reference context from which the research question originates and to highlight the research gap in order to give evidence of the contribution of the paper. So, I suggest the author(s) to rewrite the paragraph with the aim to define a logical connection between the relevance of the issue (so the necessity to observe the topic), the emersion of the gap in literature and the establishment of consequent research aims. In this perspective, the identification of the gap from literature should be adequately supported with further references.

About the theoretical sections of the paper, in general, the structure is fine, but in the content development, there are in my opinion, some limitations.

1.1 Entrepreneurial Self-Efficacy

The paragraph is a little bit confused and need to be better developed. One of the main problem, in my opinion, is about the references of previous studies about the theme. In several points, authors speak about research on the topic but do not cite any reference about.

For example:

Page 2 – Lines 46-47 authors write:  “The concept of self-efficacy is derived from Bandura's Social Learning Theory.” Please cite reference.

Page 2 - Lines 55-56, authors write: “Some scholars describe entrepreneurial self-efficacy as entrepreneurial self-confidence in specific tasks”. Please insert references.

Page 2 – Line 58-59 authors say: “Some scholars defined the concept of entrepreneurial self-efficacy from three dimensions in 2009”. Please indicate the references.

Probably the reference for this sentence is reported in the following sentences, where authors write: “The first dimension is to apply self-efficacy to specific aspects of entrepreneurial spirit. The second dimension is to emphasize the content level of self-efficacy. The third dimension is the validity of self-efficacy belief [11].

The reference number 11 is “Drnovsek M.; Wincent J.; Cardon M.S. Entrepreneurial self-efficacy and business start-up: developing a 388 multi-dimensional definition. International Journal of Entrepreneurial Behaviour & Research. 2010, 16, 389 1355–2554”.

But, if so, probably there is a mistake in one of the two sentences. The reference for the sentence in line 58/59 is 2009 or 2010?

Page 2 – Lines 61-62 authors say: “Many researchers have explored the relationship between entrepreneurial self-efficacy and other variables.” What are “other variables” and who are researchers that have investigated this relationship?

Page 2 - Line 71: “…some researches based on the Social Career Cognition Theory(SCCT)…”. Please indicate references for this theory.

In the second part of the paragraph, there is an analysis about the relationship of entrepreneurial self-efficacy and gender.  It is very difficult to understand the significance of this part in consideration of the aim of the paper.

1.2 Emotional Intelligence

Also in this paragraph there are some problems with citations and references.

Please check the citation in the sentence from line 112 on page 3 to 122 in the same page.

For example, for the following sentence: “There are currently three models of research on emotional intelligence: Ability Model represented by Mayer and Salovey, Mixed Model represented by Bar-On, and Trait Model  represented by Petrides”, authors cite a reference (number 30) of 2000.

In the following sentences, in which authors explain the three different model, all the cited references are more recent (from 2016 to 2020). So, there is a contradiction in these citations! Please check.

Page 3 - Line 141, authors say: “Career adaptability is a popular concept in organizational behavior in recent years”. Please indicate reference for carreer adaptability.

1.3. Emotional Intelligence and Entrepreneurial Self-Efficacy

The paragraph 1.3 is very poor. Authors cite some studies about the relationship between emotional intelligence and entrepreneurial self-efficacy, but about I have two doubts: The first is about the recognition made by the authors. As for the previous paragraphs, there are a lot of imprecisions and not correct references. Please check your sentences.

For example:

Lines 154-155: “At present, there are relatively few studies on the relationship between emotional intelligence and entrepreneurial self-efficacy”.  What are such studies? And what are they main results?

My second observation is about the gap that authors find. Authors say: “However, there are few studies on the emotional intelligence and entrepreneurial self-efficacy of students in Chinese vocational colleges”.

I think that so formulated and explained it is a very limited gap, that make difficult for reader to understand the contribution of the paper both from a theoretical and practical perspective. It should be also important, but authors should better explain and motivate why the exploration of the relationship between emotional intelligence and entrepreneurial self-efficacy of students in Chinese vocational colleges is important.

2. Material and Methods.

In the paragraph “participants” few information have been provide about the sample. The section should explain the sampling method used and not the result of the procedure, as the authors do. That part should be added to the sample descriptions in the results section.

In the paragraph 2.2. Instruments, authors indicate the scales used for the analysis, but no further details have been provided about variable used. A correct bibliography for the sources of the scales used in the paper is really missing. Further, Table 1 and Table 2 do not add information: tables indicate the dimensions of the two scales but do not specify the items. At the same time, there is not an appendix with this information. This limit the understanding of the paper and the reading for the reader.

A section on the control variables is missing, which should however be substantiated with adequate bibliographic references.

The Statistical Analysis section is truly superficial. All the procedures used to achieve your results should be described in detail. Authors say: “Use SPSS 25.0 software and Amos 7.0 for data entry and descriptive statistics, linear regression analysis, etc.” What does it means etc….??

Further, authors state: "we decided to perform logistic regression analysis". But they don't report any regression results in the paper. Why? It is a very serious shortcoming.

3.Results

I do not understand the meaning of tables 4 and 5, but above all it is senseless to report the Cronbach’s alpha for the three percentiles of your variables. The Cronbach's alpha is a measure of internal consistency, that is, how closely related a set of items are as a group. It is considered to be a measure of scale reliability that apply to different items of the same construct.

What are bold values ​​in the correlation table? A variable related to itself is equal to 1, while 0.93 and 0.94 appear in your table, I really don't understand these values.

Model Analysis: where is the model located? A table with the regression results is completely missing.

If that in figure 1 is a spotlight analysis (and I think so because the authors speak of slope) the moderation effect does not exist (the lines at the different levels should converge in one point). In any case, I have not seen any hypothesis of moderation ... so I still don't understand the meaning of this figure.

Overall I think this section is completely inappropriate and needs to be totally revised. The lack of the regression table is very serious.

4.Discussion

The paragraph discussion need to be further developed: in the current form, the discussion is very poor.

The paragraph about the implication of the paper require a more accurate effort: actually, theoretical implications are not provided, and practical implications of the study are just introduced.

Reviewer 3 Report

Please, consider the inclusion of

Liñán, Rodríguez-Cohard and Rueda-Cantuche (2011): Factors Affecting Entrepreneurial Intention Levels: A role for education, International Entrepreneurship and Management Journal, 7(2): 195-2018.

Ajzen, I. (1991): The theory of planned behavior. Organizational Behavior and Human Decision Processes, 50(2), 179–211.

I think theses references could enrich the analysis carried out from line 63 to 82.

I recommend to embed Conclusions section into Discussion Section.